

# Global time series and temporal mosaics of glacier surface velocities, derived from Sentinel-1 data

Peter Friedl[1], Thorsten Seehaus[1], Matthias Braun[1]

[1]Institute of Geography, Friedrich Alexander University Erlangen-Nuremberg, Erlangen, 91058, Germany

*Correspondence to*: Peter Friedl (peter.friedl@fau.de)

**Abstract**

Consistent and continuous data on glacier surface velocity are important inputs to time series analyses, numerical ice dynamic modelling and glacier mass flux computations. Since 2014, repeat-pass Synthetic Aperture Radar (SAR) data is acquired by the Sentinel-1 satellite constellation as part of ESA's (European Space Agency) Copernicus program. It enables global, near

real time-like and fully automatic processing of glacier surface velocity fields at up to 6-day temporal resolution, independent of weather conditions, season and daylight. We present a new near global data set of glacier surface velocities that comprises continuously updated scene-pair velocity fields, as well as monthly and annually averaged velocity mosaics at 200 m spatial resolution. The velocity information is derived from archived and new Sentinel-1 SAR acquisitions by applying feature and speckle tracking. The data set covers 12 major glaciated regions outside the polar ice sheets and is generated in an HPC (High

Performance Computing) environment at the University of Erlangen-Nuremberg. The velocity products are freely accessible via an interactive web portal that provides capabilities for download and simple online analyses: http://retreat.geographie.uni-erlangen.de. In this paper, we give information on the data processing and how to access the data. For the example region of Svalbard, we demonstrate the potential of our products for velocity time series analyses at very high temporal resolution and assess the quality of our velocity products by comparing them to those generated from very high resolution TerraSAR-X SAR

(Synthetic Aperture Radar) and Landsat-8 optical (ITS_LIVE, GoLIVE) data. We find that Landsat-8 and Sentinel-1 annual velocity mosaics are in an overall good agreement, but speckle tracking on Sentinel-1 6-day repeat acquisitions derives more reliable velocity measurements over featureless and slow moving areas than the optical data. Additionally, uncertainties of 12-day repeat Sentinel-1 mid-glacier scene-pair velocities are less than half ($<0.08$ m d$^{-1}$) of the uncertainties derived for 16-day repeat Landsat-8 data ($0.17 - 0.18$ m d$^{-1}$).

## 1 Introduction

Glaciers are very sensitive indicators of global climate change (Bojinski et al. 2014), since recent atmospheric warming (Allen et al. 2018) has direct or indirect influence on their mass balances (Zemp et al. 2019) and dynamics (Jiskoot 2011). Climate induced glacier change has important implications for global sea level rise (Bamber et al. 2018), freshwater availability (Huss und Hock 2018) as well as natural hazards (Moore et al. 2009). Large-scale analysis of glacier changes is performed by



observing changes in ice elevation (e.g. Braun et al. 2019; Sommer et al. 2020; Brun et al. 2017; Helm et al. 2014), ice mass
      (e.g. Gardner et al. 2013; Wouters et al. 2019), ice extent (e.g. Meier et al. 2018) or ice velocity (e.g. Dehecq et al. 2019). Ice
      velocity is determined by several factors, such as ice geometry (e.g. thickness, surface slope), physical ice properties (e.g.
      viscosity), terminal environment (land, ocean), bedrock geometry, conditions at the glacier bed (e.g. melt water availability),
      and mass balance (Jiskoot 2011). Alterations in one or more of these factors can result in long term (Dehecq et al. 2019),
seasonal (Vijay et al. 2019; Moon et al. 2014) and rapid changes in surface velocity (e.g. Bhambri et al. 2017). Ice velocity is
      a main determinant of ice discharge and hence an important variable for numerical ice dynamic modelling (Farinotti et al.
      2019) and mass balance calculations with the input-output method (e.g. Bamber und Rivera 2007; Minowa et al. 2021).
      Therefore, glacier surface velocity and its short and long-term variations have been identified as an Essential Climate Variable
      (ECV) that should be monitored on a regular and global scale (Bojinski et al. 2014).

Global ice surface velocities are currently only available from the ITS_LIVE (Gardner et al. 2018; Gardner et al. 2019) and
      the GoLIVE (Scambos et al. 2016; Fahnestock et al. 2016) data sets, which both use optical Landsat data as input for the
      velocity calculations. While ITS_LIVE currently provides annual mosaics and scene-pair velocity fields for data acquired
      between 1985 and 2018 over polar ice sheets and major glaciated regions (excluding the European Alps, the Caucasus,
      Kamchatka, Scandinavia, South Georgia and New Zealand), the GoLIVE data comprise regularly updated scene-pair velocity
fields for all of the Earth's glaciated regions from 1985 until today. However, temporal coverage and temporal resolution of
      the optical velocity data are restricted, as optical remote sensing relies on sun illumination of the Earth's surface, which leads
      to data gaps in case of cloud coverage, as well as during night and polar darkness.

      In contrast, repeat-pass Synthetic Aperture Radar (SAR) data acquired by the Sentinel-1 constellation as part of ESA's
      (European Space Agency) Copernicus program enable near real time-like and fully automatic processing of global glacier
velocities at up to 6-day temporal resolution, independent of weather conditions, season and daylight from 2014 until today.
      However, freely accessible Sentinel-1 velocity data sets to date only comprise either graphs of centerline velocities of selected
      glaciers in polar regions between 2015 – 2017 (ENVEO 2020), ice velocity fields of Pine Island Glacier in Antarctica between
      2014 – 2019 (ENVEO 2019), regularly updated 24 days composites of velocity maps of Greenland (Solgaard und Kusk 2019),
      as well as annual ice velocity mosaics of Greenland and Antarctica between 2014 and 2016 (Nagler et al. 2015). Furthermore,
there are annual velocity maps of Antarctica (Mouginot et al. 2017b; Mouginot et al. 2017a), as well as multi-year velocity
      mosaics of the Southern Patagonian Icefield (Abdel Jaber et al. 2019), Greenland (Joughin et al. 2018; Joughin et al. 2016)
      and Antarctica (Rignot et al. 2011; Mouginot et al. 2012; Rignot et al. 2017) available, which were derived from a mixture of
      Sentinel-1 and other sensors.

      Here we present a new near global data set of Sentinel-1 glacier velocities in 12 regions outside the polar ice sheets (Fig. 1)
that comprises scene-pair velocity fields, as well as monthly and annual velocity mosaics derived from applying intensity offset
      tracking on both archived (since 2014) and the continuous stream of new acquisitions. We describe the procedures of data
      generation in detail, give information on how to access the data, demonstrate the capabilities of our products for velocity time
      series analyses at very high temporal resolution and provide a comprehensive comparison of our data set with velocity products





generated from very high resolution TerraSAR-X radar and Landsat-8 optical (ITS_LIVE, GoLIVE) data. In this paper, we

demonstrate the performance of our processing on Svalbard as an example region, because it includes glaciers that are characterized by a broad variety of sizes, different velocity magnitudes and seasonal velocity patterns. On Svalbard, there are also ice caps and ice fields with almost featureless surfaces, as well as surging glaciers that are prone to very rapid and strong accelerations.

## 2 Data and Methods

### 2.1 Sentinel-1 intensity offset tracking

All processing of our glacier surface velocities is conducted in FAU's (Friedrich Alexander University) HPC (High Performance Computing) environment, currently consisting of 246 compute nodes that have a total amount of 984 CPUs and 6192 GB RAM. Our main input data are consecutive pairs of single or dual polarized Sentinel-1 SLC (Single Look Complex) SAR (Synthetic Aperture Radar) images, acquired over 12 glaciated regions outside the Antarctic and Greenland ice sheets

(Fig. 1). SLC images contain both phase and amplitude information. ESA's (European Space Agency) Sentinel-1 constellation currently consists of two satellites, Sentinel-1A (launched on 3 April 2014) and Sentinel-1B (launched on 25 April 2016), both carrying a C-Band SAR (Synthetic Aperture Radar) sensor operating at a frequency of 5.405 GHz (Geudtner et al. 2014). Each satellite has an exact revisit time of 12 days. A minimum repeat cycle of 6 days is achieved in regions where both satellites acquire data. Such regions currently comprise the Antarctic Peninsula, Greenland, Arctic Canada and some selected European

sites, including Svalbard. Due to the active measuring principle and since radar signals penetrate through clouds, suitable data are available all year round, (polar-) night and day, and under all weather conditions. Sentinel-1 has four different imaging modes with different resolutions and spatial coverages (Torres et al. 2012). We use data recorded in IW (Interferometric Wide swath) mode at a pixel spacing of ~14 m in azimuth (az) and ~3 m in range (r), and a spatial coverage of ~250 x 250 km. Data are available in single (HH or VV) or dual polarization (HH+HV or VV+VH), of which we only use the HH or VV channels.

Archived SAR data were automatically downloaded from the ASF (Alaska Satellite Facility) DAAC (Distributed Active Archive Center) and our database is routinely updated with new imagery from the same source. New data is available within three days of acquisition, which allows for near real time-like velocity processing. Over Svalbard, data coverage starts in January 2015. By January 2021, we had processed roughly 110.000 Sentinel-1 SLC scenes (~450 TB) for all 12 regions of interest and more than 2.100 scenes (> 8 TB) for Svalbard alone. For the following years, we estimate the yearly amount of

processed data to be ~ 24.000 scenes (~ 100 TB).

Sentinel-1 IW imagery is acquired using the TOPS (Terrain Observation with Progressive Scans in azimuth) technique (de Zan und Monti Guarnieri 2006; Geudtner et al. 2014). TOPS allows for larger swath widths than the classic strip map mode by steering the antenna back and forth in both the azimuth and the range direction, but the achievable azimuth resolution is lower due to a reduced target dwell time in azimuth (Geudtner et al. 2014). Sentinel-1 IW SLC images usually consist of 3

sub-swaths per polarization channel, each of them divided into 9–10 single bursts, whereas each burst is affected by a linear





azimuth phase ramp due to the rapid change of the azimuth antenna pointing. The differences between the Sentinel-1 TOPS and normal strip-map mode acquisitions require some additional processing steps.

First, we update the state vectors of the Sentinel-1 IW SLC images using recalculated POD (Precise Orbit Determination service) precise orbit ephemerides information that are available within three weeks after acquisition to assure highest
geolocation accuracy (5 cm 3D 1-sigma RMS). This and the temporal separation between the images lead to a time lag of regularly produced velocity fields of about 3–6 weeks. However, switching to less precise (10 cm 2D 1-sigma RMS) POD Restituted Orbit data that is available within 3 hours after acquisition, is possible if a more near real time-like processing is required (e.g. to establish an early warning system). We then precisely coregister consecutive pairs of overlapping images taken at the same path and frame using an iterative three-step coregistration procedure, tailored to the special requirements of
Sentinel-1 TOPS data (Wegmüller et al. 2016). Choosing the proper time separation between the images is a trade-off between minimizing the measuring error (Equ. 2) and maximizing the temporal resolution of the velocity time series, considering the expected surface displacement, surface characteristics and the data availability in the respective area. Depending on the region, we selected a minimum time separation of 6–48 days, whereas temporal baselines of up to 96 days are allowed, if no other data is available (Table S1). For Svalbard the minimum temporal baseline was 6 days for data from 2016 onwards and 12 days
for data prior to 2016, respectively. The time stamp of the resulting products is taken as the mean date of the corresponding image pair. The coregistration consists of 1) a rough coregistration based on the information contained in the orbit parameter file, 2) an iterative intensity cross-correlation offset estimation until the azimuth correction determined is <0.01 pixel or until 5 iterations are reached and 3) an iterative spectral diversity method if phase coherence is retained (Scheiber und Moreira 2000). The latter minimizes residual phase offsets between the burst-overlap regions, until the azimuth correction determined
is <0.001 pixel or until 5 iterations are reached. In order to facilitate oversampling during tracking, the bursts of the master image of each processing pair are corrected for their azimuth phase ramps (de-ramping) and the derived correction function is then applied to the bursts of the slave scene (Miranda 2017; Wegmüller et al. 2016). After de-ramping, the bursts are mosaicked and a well-established intensity offset tracking algorithm implemented in the GAMMA software package is applied, which uses a moving window approach to determine normalized cross correlation peaks between patches of the master and the slave
intensity image in order to derive azimuth and slant range displacement (Strozzi et al. 2002; Wegmüller et al. 2016; Friedl et al. 2018; Wendleder et al. 2018; Seehaus et al. 2018). The technique is based on tracking persistent patterns of intensity values in both images, which are either formed by surface features such as crevasses (feature tracking) or correlated radar speckle (speckle tracking). In contrast to optical data, the latter enables radar data to derive more reliable tracking results in slow moving accumulation areas or over large ice caps with featureless and smooth surfaces. However, since speckle tracking
requires phase coherence, its application is often restricted to winter acquisitions when there is no surface melt and to regions where 6 day-repeat data is available and where surface velocities are low (i.e. accumulation areas, ice cap interiors) (Fig.2). Tracking window sizes need to be selected according to the expected displacement, the size of the glaciers and the size of the features to be tracked. In the case of Svalbard we use a tracking window size of 250 r x 50 az pixels and the step size is chosen to be 50 x 10 pixels in the range and azimuth direction. Table S1 summarizes the tracking parameters for each region. During





tracking, invalid displacement measurements are rejected if their Cross-Correlation Peak coefficient (CCP) is below 0.08, whereas a CCP of 1 indicates perfect cross correlation. However, since this procedure just removes very bad blunders, further filtering is applied during post-processing (Sect. 2.2).

The raw displacement fields are converted from slant range into ground range by means of the local incidence angles, computed from the topographic information of a Digital Elevation Model (DEM). Additionally, the DEM serves as a reference for

geocoding and orthorectification, as well as for the removal of velocity results affected by topographic distortions in the SAR signal (layover and shadow). For regions between 60 ° N and 56 ° S, we use the void-filled 3 arc second (~90 m) global NASA (North American Space Administration) SRTM (Shuttle Radar Topography Mission) DEM Version 3 (Farr et al. 2007; NASA JPL 2013) and for all other regions the DLR (German Aerospace Center) global TanDEM-X DEM at 3 arc second resolution (Wessel et al. 2018) as a reference DEM. The resulting intermediate velocity products are UTM-geocoded and orthorectified

rasters in GeoTIFF format, resampled to a spatial resolution of 200 m. The rasters comprise horizontal surface displacements (m d$^{-1}$) in range and azimuth direction (i.e. relative to the sensor's flight path), the magnitude of the velocity vector, the CCP and CCS (Cross Correlation function Standard deviation) values, as well as the angle of displacement relative to the sensor's heading direction and the angle of displacement relative to true north.

## 2.2 Post-Processing and error estimation of scene-pair velocity fields

Our post-processing procedure consists of additional filtering and correction for remaining coregistration errors. For filtering we apply a three-step approach of Lüttig et al. (2017) to the intermediate azimuth- and range velocity fields. All other intermediate velocity products (i.e. magnitude and angles of displacement) are masked accordingly. It was shown that the filtering method removes up to more than 99 % of erroneous data points, while keeping a maximum of valid velocity measurements (Lüttig et al. 2017). In a first step, velocity fields are recursively divided into segments that are smooth within

themselves, by comparing the velocity differences between random seed points $p$ and their neighbors $n$ with a threshold $t$:

$$t = e_{const} + \Delta v \cdot w \qquad (1)$$

where $e_{const}$ is a constant error computed as the square root of the quadratic sum of the errors of the offset tracking algorithm (Eq. 2) and the coregistration (conservatively assumed to be a constant of 0.08 m d$^{-1}$) multiplied by a factor of 0.3, $\Delta v$ is the difference between $p$ and $n$ in an a-priori reference velocity field and $w$ is a variable factor accounting for possible temporal changes between the a-priori field and the actual data. While Lüttig et al. (2017) propose $w = 1.5$ for regions of relatively

stable velocities, we selected $w = 3$ in order to account for the strong seasonal velocity signals and surging behavior of many glaciers in Svalbard. Data points where $p - n$ exceeds $t$ in one of the two directions, are not assigned to the corresponding segment and segments that contain less than 8 measurements are removed.

To assure that possible blunders related to the sensor's characteristics or the processing procedure are removed properly, the a-priori velocity information should be selected so that it is independent from the data to be filtered. Hence, we use annual

mean surface velocity mosaics at a spatial resolution of 240 m, that were generated by applying feature tracking to optical Landsat-8 imagery within the ITS_LIVE project (Gardner et al. 2018; Gardner et al. 2019). In order to make our range and





azimuth velocities comparable to their x (East-West) and y (North-South) velocities, we transform our range and azimuth pixel values to x and y values relative to the projection of the ITS_LIVE data. Our filtered x and y velocities are then used to mask the original range and azimuth displacement values. If possible, the Landsat reference velocities are selected to match the year of the Sentinel-1 data. Velocity fields dated after 2018 are preliminarily filtered using the latest available ITS_LIVE data from 2018. For such data, filtering may be repeated, once mosaics of the corresponding year are available. For pixels or regions where the ITS_LIVE reference velocities have gaps or are not available, the first filtering step is skipped.

In the second filtering step, the medians of the remaining range and azimuth velocities as well as the corresponding standard deviations are calculated for a 5 x 5 pixels moving window. All measurements where the difference between the velocity and the median exceeds 3 times the standard deviation in at least one of the velocity components, are discarded. Similarly, in the third filtering step, a 5 x 5 pixels moving window is used to remove range and azimuth velocity components that have a difference to the window's mean direction of more than 3 times the window's standard deviation. Additionally, all data points are removed that have a direction difference of more than 20° to more than 4 neighboring points or that have less than 2 neighbors within the window. Figure 2 shows examples of the filtering results for velocity fields over two different regions in Svalbard.

Although our coregistration procedure aims for high precision, remaining coregistration errors are inevitable. Usually, absolute coregistration errors of the velocity's magnitude are around or well below 0.01 m d$^{-1}$, but in some cases they can exceed 0.05 m d$^{-1}$, especially if scene pairs do not cover a sufficient amount of stable ground. Because of the higher range resolution of the Sentinel-1 IW SLC data, coregistration errors are frequently up to one order of magnitude lower in range than in azimuth. Assuming that the coregistration bias is a uniform shift in the range- and azimuth-direction over the entire velocity field, we determine the bias by calculating the median of the filtered range- and the azimuth-velocities over stable ground that is not covered by ice or water (Fig. 3). For this we use a mask, which we generated by subtracting water bodies contained in the HydroLAKES data set (Messager et al. 2016) and glaciers contained in the Randolph Glacier Inventory (RGI) 6.0 (RGI Consortium, 2017) from an OpenStreetMap-based land polygon data set (https://osmdata.openstreetmap.de/data/land-polygons.html). We correct the filtered range- and azimuth-velocity fields by adding or subtracting the determined coregistration biases and recalculate the magnitude and the angles of displacement. While in most cases the applied absolute corrections are very small (around or well below 0.01 m d$^{-1}$), the procedure significantly improves the measurement quality in regions that are difficult for coregistration (e.g. ice caps with a small amount of stable ground) (Fig. 2).

Assuming that the correction successfully removed existing coregistration errors, we estimate the remaining velocity error to be a function of the tracking accuracy of 0.1 pixel, the pixel size $ps_r$ and $ps_{az}$ in meters in each direction, as well as the temporal baseline $tb$ of the image pair in days (Mouginot et al. 2017b):

$$e = \sqrt{\left(\frac{0.1 * ps_r}{tb}\right)^2 + \left(\frac{0.1 * ps_{az}}{tb}\right)^2} \qquad (2)$$





This results in theoretical errors of 0.24 m d$^{-1}$ and 0.12 m d$^{-1}$ for Sentinel-1 IW data with a pixel size of 3 x 14 m, acquired at the typical repeat cycles of 6 and 12 days, respectively. However, the results of an inter-comparison experiment between Sentinel-1 and TerraSAR-X velocity measurements, as well as experiments of a similar kind conducted by others (Sect. 3.2)

suggest that in reality the uncertainty of mid-glacier surface velocities generated from 12-day Sentinel-1 IW repeat imagery is lower (~0.08 m d$^{-1}$).

## 2.3 Annual and monthly velocity mosaics

For all regions (Fig. 1), we calculate annual and monthly mosaics from all post-processed velocity products that have a time stamp between 1 January–31 December of a year and between the first and the last day of a month, respectively. New annual

and monthly mosaics become regularly available with a time lag of 2 months.

Before mosaicking, the UTM-projected scene-pair velocity products are reprojected to a common coordinate reference system (e.g. NSIDC Sea Ice Polar Stereographic North in case of Svalbard) and range and azimuth displacement values are transformed to x (East-West) and y (North-South) velocity components relative to true north, in order to allow for direct comparability. If the number of measurements per pixel is >2, we calculate the median and the standard deviation for both the

x and y displacements for each pixel. Measurements that have an absolute difference to the median of more than two times the standard deviation in at least one of the two directions are removed and not considered for further processing (Mouginot et al. 2017b). If there is only one measurement per pixel, we keep the measurement as it is. Taking the SNR (computed as SNR=CCP/CCS) as weights, we then calculate the weighted mean, weighted standard deviation and the weighted standard error for the x and y velocity components, as well as the magnitude of the velocity for each pixel. Additionally, we derive the

weighted means of the acquisition date (days since 1 January 1900) and the time separation between the images, the displacement angle relative to true north (based on the weighted means of the x and y velocity components), as well as the number of measurements per pixel. In regions where glaciated areas are separated by large ice-free areas, the velocity mosaic products are clipped according to an ice mask that we generated by applying a 10 km buffer to the RGI 6.0 glacier inventory.

## 2.4 Naming convention and data availability

All scene-pair glacier velocity fields and mosaics that we have produced so far and will regularly produce in the future, are made freely available via a bilingual (German, English) web portal that can be accessed at http://retreat.geographie.uni-erlangen.de. In addition to a standard spatial search and download function, the portal also offers the possibility to the users to generate and download their own velocity time series based on individually drawn glacier profiles. Furthermore, the subset of Svalbard used in this paper, is separately available at GFZ (German Research Centre for Geosciences) Potsdam Data Services

(see data availability section).

Scene-pair glacier velocity products and mosaics follow the naming conventions shown in Table S2. Both product types are accompanied by a metadata file that contains information on e.g. the input- and auxiliary data, tracking parameters, velocity



error and applied correction factors (in case of scene-pair velocity fields), as well as the number of velocity fields that were used in the averaging process (in case of mosaics).

## 2.5 Comparison of Sentinel-1 and Landsat-8 scene-pair velocity fields with TerraSAR-X

In order to assess the quality of our Senintel-1 measurements, we compare them, together with Landsat-8 scene-pair velocities from the ITS_LIVE (Gardner et al. 2018; Gardner et al. 2019) and GoLIVE (Scambos et al. 2016; Fahnestock et al. 2016) data sets, with velocity products that we generated from TerraSAR-X radar imagery of much higher resolution and precision (Paul et al. 2017; Strozzi et al. 2017; Nagler et al. 2015).

To generate the TerraSAR-X velocities, we applied intensity offset tracking (Sect. 2.1) to 11-day repeat pass Strip Map (SM) acquisitions at a spatial resolution of ~3 m, using a 128 x 128 pixels window size and a step size of 25 x 25 pixels. The velocity fields were orthorectified, filtered and corrected as described in Sect. 2.1. However, the average absolute correction factors determined over stable ground were very small (<0.005 m d$^{-1}$), reflecting the high coregistration accuracy of the TerraSAR-X velocity products. Assuming that the tracking accuracy is 0.1 pixel, the TerraSAR-X velocities have formal errors of 0.04 m d$^{-1}$ (Eq. 2).

All data sets were chosen to offer a good balance between spatial coverage and temporal overlap. Nevertheless, slightly different imaging intervals were inevitable (see Table 1 for acquisition dates). From the ITS_LIVE and GoLIVE data sets, we selected velocity fields with a temporal baseline of 16 days, in order to best match the repeat intervals of the Sentinel-1 (12 days) and the TerraSAR-X (11 days) data. To assure the highest quality of the Landsat-8 velocities, we only selected products that were generated from consistently georegistered Landsat-8 Tier 1 data (Young et al. 2017). Furthermore, we made sure that the ITS_LIVE and GoLIVE velocity fields were derived from the identical input imagery. The Landsat-8 velocity products have spatial resolutions of 240 m (ITS_LIVE) and 300 m (GoLIVE), and were produced using different feature tracking procedures, which are described in more detail in Gardner et al. (2018) and Fahnestock et al. (2016). Both processing schemes take preprocessed panchromatic Landsat-8 images at 15 m pixel size as input and involve masking of unreliable measurements based on a cross correlation peak threshold and neighborhood similarity, as well as correction for geolocation/coregistration errors based on stable ground velocities. The theoretical error of the Landsat-8 ice flow measurements is 0.13 m d$^{-1}$ under the assumption of a measurement precision in ice flow of 0.1 pixels (Eq. 2), but it may be larger depending on the successful correction of geolocation/coregistration errors (Fahnestock et al. 2016). Additionally to the velocity magnitude, we analyze the displacement angles associated with the surface velocities, as they are important inputs to ice flux/mass balance calculations and numeric ice modelling. For this, we computed the displacement angles relative to true north for all input data sets.

Velocities and displacement angles were compared for four different glaciers in Svalbard: Kronebreen, Negribreen, Tunabreen and Strongbreen (Fig. 4). In contrast to the time series analysis in Sect. 3.1.1, we did not consider Austfonna Basin 3 and Bodleybreen, since no TerraSAR-X data was available at these sites for the period 2015–2020. Additionally, due to its dependency on sunlight, the availability of 16 day Landsat-8 velocities was restricted to the summertime. This resulted in an





unavailability of Landsat-8 data over Negribreen during winter 2015, when TerraSAR-X and Sentinel-1 velocities have a
temporal overlap.

For each glacier we extracted ice surface velocities and displacement angles along the centerlines of Nuth et al. (2013), which
we clipped according to the data coverage of the velocity fields (Figs. 6 and 7). Similar to Strozzi et al. (2017), we then
calculated the uncertainty as the median and the NMAD (Normalized Median Absolute Deviation) of the differences between

the "true" TerraSAR-X velocities and displacement angles, and the corresponding Sentinel-1 and Landsat-8 measurements
over a) regions close to the glacier's calving fronts and shear zones, b) mid-glacier regions far away from the calving fronts
and shear zones and c) regions of stable ground (Fig. 6 and Table 1). We did not calculate mean differences and standard
deviations as originally proposed by Paul et al. (2017) and Strozzi et al. (2017), because both measures are very sensitive to
single outliers, which would distort the statistics especially for the GoLIVE data, which contain sporadic erroneous pixels of

very high (up to > 30 m d$^{-1}$) velocities (Fig. 6). We primarily attribute discrepancies between the TerraSAR-X and the other
velocity data sets to uncertainty in the Sentinel-1 and Landsat-8 measurements. However, differences in the representativeness
of the displacements to the "true" displacement and temporal velocity variations between the slightly different acquisition
dates are influencing factors, too (Paul et al. 2017).

## 3 Results and Discussion

### 3.1 Velocity time series from Sentinel-1 scene-pair velocity fields at very high temporal resolution on Svalbard

In order to demonstrate the capabilities of our dataset for high temporal resolution time series analyses, we extracted surface
velocities from all available post-processed Sentinel-1 scene-pair velocity fields (1 January 2015–30 November 2020) over 6
glaciers in Svalbard. For each glacier, velocity values were computed as the median displacement within a 500 m buffer around
three points along the glacier's centerline (Fig. 4). The centerlines were taken from the glacier inventory of Svalbard, GI$_{00s}$,

by Nuth et al. (2013), whereas the centerline of Negribreen was adjusted according to a significant change in the front's flow
direction that happened around 2010 (Haga et al. 2020). As a result, we got very dense, complete and consistent velocity time
series for all six glaciers that document distinct patterns of short-term seasonal velocity variations, glacier surges and longer-
term velocity trends over the last five years (Fig. 5). The data density of the time series increased in 2017, when more
acquisitions from both Senintel-1A/B became available over Svalbard.

Following a stepwise frontal acceleration of Austfonna Basin 3 between 2008–2012 from ~2 m d$^{-1}$ to ~4 m d$^{-1}$ and a surge in
2012/2013 with maximum velocities of ~19 m d$^{-1}$ (Dunse et al. 2015), our Sentinel-1 time series for 2015–2020 reveals an
ongoing gradual slowdown of the glacier, overlain by a seasonal cycle of summer (July-August) acceleration (Fig. 5a). Summer
peak velocities at the glacier front decreased from ~10.5 m d$^{-1}$ in 2015 to ~8 m d$^{-1}$ in 2019, but were ~9.5 m d$^{-1}$ in 2020, which
is still far away from the pre-surge level of ~2 m d$^{-1}$ prior to 2012. Our results are in very good agreement with the numbers

reported by Strozzi et al. (2017) for a Sentinel-1 velocity time series of Basin 3 covering 2015–2017. The characteristic of a
long surge duration (5–tens of years) relative to surges in other regions (typically 1–4 years), often including multi-year



acceleration and deceleration phases, is considered typical for surging glaciers in Svalbard (Dowdeswell et al. 1991; Murray et al. 2003a; Murray et al. 2003b).

Our time series also captures the recent surge of Negribreen at high temporal detail (Fig. 5d). The surge was initiated by a
stepwise increase in frontal velocity over the 2015 melt season from <1 m d$^{-1}$ to ~3 m d$^{-1}$, followed by slight slow-down during winter 2015, rapid acceleration during summer and autumn 2016, an interphase of almost constant frontal velocities of ~14– 16 m d$^{-1}$ during spring 2017, a final maximum peak of ~24 m d$^{-1}$ in the melt-season of 2017 and a period of ongoing gradual deceleration with typical summer acceleration peaks. The course of velocity as well as the measured velocity magnitude values, are in very good agreement with recent measurements from independent multi-sensor (Haga et al. 2020) and Sentinel-1 velocity
time series (Strozzi et al. 2017). Furthermore, we are able to demonstrate that our tracking and filtering procedures allow to measure short-time events of exceptional high surface velocities.

Different to the surges of Austfonna Basin 3 and Negribreen, the marine terminating glaciers Tunabreen (Fig. 5e) and Strongbreen (Fig. 5f) showed no phase of multi-year acceleration prior to the main rapid acceleration phase. On Tunabreen, the surge lasted only 2 years (autumn 2016 – autumn 2018) and terminated with a more abrupt deceleration rather than a
protracted slowdown, which is more similar to surges in other mountain regions such as Alaska or the Karakoram. However, the maximum velocities of ~6 m d$^{-1}$ during the melt season in 2017 are relatively low in comparison to surging glaciers in other regions (Murray et al. 2003b). Additionally, at Tunabreen we do not find any clear seasonal velocity pattern during the pre- and post-surge phases. Whereas the velocity time series of Negribreen and Tunabreen show that the surges initiated in the lower areas of the glacier and then spread upstream, for Strongbreen the time series reveals that the surge started from the
upper areas, followed by a surge front of fast-moving ice propagating down the glacier. The latter is similar to what is reported for surges of land-terminating glaciers in Svalbard (Hagen 1987; Murray et al. 1998), Alaska (Kamb et al. 1985) and the Karakoram (Quincey et al. 2011).

For Bodleybreen (Fig. 5b), our velocity time series documents a characteristic seasonal cycle of relatively stable velocities during winter, rapid deceleration in late spring (Mai/June) with minimum frontal velocities of ~0.5 m d$^{-1}$ in August/September
followed by a rapid acceleration that re-gains a winter velocity level of ~2 m d$^{-1}$ until December. The velocity pattern is very similar to the "type-3" pattern of glaciers in Greenland, identified by Moon et al. (2014) and Vijay et al. (2019). This velocity pattern is associated with a seasonal switch between active (efficient) and inactive (inefficient) subglacial meltwater drainage channels: Active subglacial drainage channels develop quickly close to the onset of the melt season and meltwater is efficiently drained, causing both rapid subglacial water pressure and ice velocity decrease in summer. During autumn and winter, drainage
channels close and become inactive, likely due to viscous deformation, leading to re-acceleration in response to water pressure buildup caused by different possible water sources, such as e.g. basal meltwater, infiltrating ocean water, summer meltwater retained in the firn and ice body (Vijay et al. 2019) and rainfall (Schellenberger et al. 2015).

A different seasonal pattern is revealed for Kronebreen (Fig. 5c), where periods of relatively constant velocities during winter and spring are interrupted by phases of significant acceleration starting in Mai and peaking in July, followed by a significant
drop in velocity that reaches its minimum in late summer/autumn and subsequent re-acceleration to the original winter





velocities. This seasonal pattern is confirmed by previous GPS (Global Positioning System) and SAR measurements (Schellenberger et al. 2015) and its characteristic is in between the Greenland glacier "type-1" and "type-3" patterns suggested by Moon et al. (2014) and Vijay et al. (2019). Here, the prominent early summer speedup is likely linked to increasing subglacial water pressure in response to surface melt input that cannot be routed by the still inefficient drainage system. As

soon as the drainage channels become active, the subglacial water pressure and the velocity drop. This is followed by re-acceleration, once the drainage system becomes inactive (possibly due to viscous deformation) and subglacial water pressure rises due to water input from different sources. Close to the front, we see an overlaying long-term velocity cycle of acceleration from 2015–2017 and deceleration since winter 2017, additionally to the general seasonal pattern. In an earlier study, general acceleration between 2011 and 2012/2013 was correlated to a reduction in backstress caused by a retreat of the glacier front

(Schellenberger et al. 2015), as it is observed for many calving glaciers all over the world (e.g. Sakakibara und Sugiyama 2018; Carr et al. 2017; Sakakibara und Sugiyama 2014). However, Sentinel-1 images acquired between 2015 and 2020 suggest that acceleration during 2015–2017 falls into a period of relatively stable front position, whereas a frontal retreat of ~800 m is documented for the deceleration phase between 2017 and 2020. While this deviation from the worldwide acceleration trend of retreating calving glaciers is an interesting topic to investigate in detail, it is beyond the purpose of this study.

Overall, we find that our data set provides very dense, continuous and consistent time series of ice velocities at very high temporal resolution for glaciers of different characteristics all year round. This allows for analyses of short and long-term glacier velocity fluctuations at new unprecedented detail.

## 3.2 Comparison of Sentinel-1 and Landsat-8 scene-pair velocity fields with TerraSAR-X

Figure 6 displays the different velocity fields used for the inter-comparison experiment. The velocity fields differ in their
spatial coverage, with TerraSAR-X having only few data gaps over flowing ice. However, directional filtering of the TerraSAR-X data removed more pixels over stable and very slow moving/stagnant ice areas in comparison to Sentinel-1. This is because accuracy of the TerraSAR-X data is better and velocities in these areas are consistently closer to zero, which in turn leads to a larger variability of neighboring displacement angles (Lüttig et al. 2017). Although Sentinel-1 data has more gaps over flowing ice than the TerraSAR-X data, the maps show a good agreement of both data sets. Coverage of the ITS_LIVE
data is denser than in the other data sets, but most velocities over stable ground and slow moving ice are quite high (up to >0.7 m d$^{-1}$), which is visible as yellowish color coding in the maps. In the GoLIVE data, more measurements were filtered out than in the ITS_LIVE data and velocities in slow moving areas appear to be lower, but single erroneous (red) pixels of very high velocities are still visible. Interestingly, while ITS_LIVE has a good coverage over Strongbreen (Fig. 6d$_3$), GoLIVE has almost no valid data points (Fig. 6d$_4$), although both data sets used the same input imagery.

Figure 7 shows that for all four glaciers, the Sentinel-1 and TerraSAR-X velocity profiles are in very good agreement, both in slow and fast flowing regions. However, TerraSAR-X velocities are smoother than the other velocity data sets. Velocity discrepancies between TerraSAR-X and the optical data sets are generally larger than those between TerraSAR-X and Sentinel-1, with ITS_LIVE overestimating the velocity by up to >0.5 m d$^{-1}$ in slow moving regions (Fig. 6c and Fig. 6d). Nevertheless,





a general clear pattern of over- or underestimation of the velocities is not detectable for the optical data. It is noticeable that

although ITS_LIVE and GoLIVE use the same input data, there are also considerable differences between both data sets, which reflects differences in the processing strategies and the applied geolocation/coregistration correction.

If looking at the displacement angle profiles in Fig. 7, there is a good match between the Sentinel-1 and the TerraSAR-X measurements, especially in regions that flow faster than ~0.5 m d$^{-1}$. Although there is a good agreement between the optical and the TerraSAR-X displacement angles in parts of these regions, the discrepancies are generally larger than for Senintel-1,

which is visible in Fig. 7a and in Fig. 7c for measurements between 0–7 km to the front. Here, the velocity differences of the optical data, which are at least partly a consequence of the applied geolocation/coregistration correction, likely translate into deviations of the displacement angle. Also for slow ice velocities between ~0.1–0.5 m d$^{-1}$, TerraSAR-X displacement angles are relatively consistent (Fig. 7b and c), which is a consequence of the very high resolution and accuracy of the TerraSAR-X data. In contrast, the lower resolution of the Sentinel-1 and the Landsat-8 imagery results in larger variabilities of their

displacement angles over such slow moving ice regions. However, for the almost stagnant front of Strongbreen (Fig. 7d, 0– 2.5 km to front), the variability in displacement angles is high for all four data sets.

Table 1 contains the median and the NMAD values of the differences between TerraSAR-X and the other data sets for each of the four glaciers, as well as the overall average of these measurements. However, not all measurements (indicated with brackets in Table 1) were taken into account for the calculation of the overall average values for several reasons: a) Displacement angles

over Negribreen were only available for Sentinel-1 and their discrepancies to TerraSAR-X are inevitably large due to the slow velocities of the glacier. Consideration of these quite large differences for Sentinel-1 only, would have biased the overall average. b) Displacement angles over the front and shear zones of Strongbreen are not meaningful, as the ice there is almost stagnant. For the same reason we did not calculate the median differences and the NMAD for displacement angles over stable ground. c) There were too few valid measurements in the GoLIVE velocity map over Strongbreen. Minimum differences to

TerraSAR-X highlighted in Table 1 as bold text, illustrate that Sentinel-1 outperforms both Landsat-8 data sets in most of the cases.

The overall average of the median velocity difference and the NMAD between Sentinel-1 and TerraSAR-X were -0.005 m d$^{-1}$ and 0.153 m d$^{-1}$ for areas close to the calving front, respectively. However, while median velocity differences were pretty low and ranged between -0.067 m d$^{-1}$ and 0.061 m d$^{-1}$, the NMAD ranged from 0.04 m d$^{-1}$ over the almost stagnant front of

Strongbreen to 0.262 m d$^{-1}$ over the fast flowing front of Tunabreen. Our average values are in good agreement with the results of a similar comparison experiment over Svalbard between Sentinel-1 and Radarsat-2 WUF (Wide Ultra-Fine, ~3 m spatial resolution) by Strozzi et al. (2017), who report an overall average velocity difference of 17 m a$^{-1}$ (0.047 m d$^{-1}$) and a standard deviation of 64 m a$^{-1}$ (0.175 m d$^{-1}$) over frontal areas of Austfonna Basin-2, Basin-3 and Stonebreen. While velocity differences of the GoLIVE data were quite similar to those of Setinel-1 over frontal areas, median ITS_LIVE velocities over the fast

flowing Tunabreen front were 0.259 m d$^{-1}$ lower than the TerraSAR-X velocities and 0.335 m d$^{-1}$ higher over the very slow flowing frontal part of Strongbreen. For both Landsat-8 velocity data sets, the overall average of the NMAD values over the glacier fronts and shear zones were higher than for the Sentinel-1 data and were 0.281 m d$^{-1}$ for ITS_LIVE and 0.254 m d$^{-1}$ for



GoLIVE, respectively. In general, uncertainty is larger at fast flowing calving fronts, because here spatial and temporal variability of ice surface velocity is large and fast-moving spots at the glacier's front are mixed with areas of much lower

velocity in the relatively large tracking windows used for image correlation (Strozzi et al. 2017).

Different to surface velocities, the overall average of the NMAD values of the displacement angles over the fast flowing fronts was low (<7°) and similar (between 4.07° and 6.87°) for all data sets. However, while the maximum median difference between TerraSAR-X and Sentinel-1 was only -3.81°, median differences between TerraSAR-X and ITS_LIVE and GoLIVE were up to -10.87° and -11.83°, respectively.

For mid-glacier areas, where velocities are between ~0.5–1 m d$^{-1}$, the overall averages of the median difference and the NMAD between the TerraSAR-X and Sentinel-1 velocity measurements were 0.003 m d$^{-1}$ and 0.079 m d$^{-1}$. These values are again very well in line with the results of the velocity comparison experiment by Strozzi et al. (2017), who found average differences between Radarsat-2 and Sentinel-1 12-day velocity records of 17 m a$^{-1}$ (0.047 m d$^{-1}$) and a standard deviation of 26 m a$^{-1}$ (0.071 m d$^{-1}$) over three different mid-glacier regions in Svalbard. Additionally, the standard deviation of Strozzi et al. (2017)

and our NMAD value are similar to an uncertainty of 0.068 m d$^{-1}$, derived for slow moving areas (0.1–0.5 m d$^{-1}$) on the Greenland west coast based on the RMSE (Root Mean Square Error) between Sentinel-1 12-day repeat and TerraSAR-X 11-day repeat measurements (Nagler et al. 2015).

However, while the overall averages of the median velocity differences between the Landsat-8 data sets and TerraSAR-X of mid-glacier regions where also low (0.014 m d$^{-1}$ for ITS_LIVE and 0.086 m d$^{-1}$ for GoLIVE), the overall averages of the

NMADs were inherently larger than for Sentinel-1 and amounted to 0.182 m d$^{-1}$ for ITS_LIVE and 0.177 m d$^{-1}$ for GoLIVE. Similarly, the overall averages of the median difference and the NMAD of the displacement angles were just -2.37°/8.66 ° for Sentinel-1, but 2.93°/45.63° and -27.69°/23.59° for Landsat-8 ITS_LIVE and GoLIVE, respectively.

Our statistical measurements over stable terrain are consistent with our observations over mid-glacier areas. While the overall average of the median difference between Sentinel-1 and TerraSAR-X is only -0.037 m d$^{-1}$ and the overall average of the

NMAD is 0.036 m d$^{-1}$, the overall averages of the median differences and NMADs of ITS_LIVE and GoLIVE are -0.239 m d$^{-1}$/0.08 m d$^{-1}$ and -0.191 m d$^{-1}$/ 0.115 m d$^{-1}$, respectively. Since TerraSAR-X velocities over stable ground are pretty close to zero, the differential values are similar to median or mean velocities frequently measured for Sentinel-1 and Landsat-8 over stable ground. Quite high mean velocities and standard deviations of Landsat-8 GoLIVE 16-day repeat velocities over stable terrain of ~0.1–~1.0 m d$^{-1}$ and ~0.1–~0.7 m d$^{-1}$, respectively, were also observed by Haga et al. (2020).

Based on the overall average of the NMAD over mid-glacier areas of 0.079 m d$^{-1}$, we estimate the uncertainty of Sentinel-1 12-day repeat velocities to be <0.08 m d$^{-1}$ over glacier regions upstream of the calving front, which is within the uncertainty range of 20–30 m a$^{-1}$ (0.05–0.08 m d$^{-1}$) estimated for mid-glacier regions by Strozzi et al. (2017) and similar to the uncertainty of 0.068 m d$^{-1}$, reported by Nagler et al. (2015). This empirical value is lower than the theoretical velocity error of 0.12 m d$^{-1}$ for velocity products derived from Sentinel-1 data with a 12-day time interval, assuming a tracking uncertainty of 0.1 pixels

(Sect.2.2). However, our experiment shows that uncertainties of 16-day repeat Landsat-8 velocity data are more than twice as high as for Sentinel-1 (0.17 – 0.18 m d$^{-1}$) in mid-glacier areas. This is more than the theoretical velocity error of 0.13 m d$^{-1}$





(Equ. 2) and suggests that additionally to the tracking uncertainty, an error is introduced by an imperfect geolocation/coregistration correction (Fahnestock et al. 2016). Nevertheless, the uncertainties of the scene-pair velocities are substantially reduced, if input data with much larger temporal baselines is used. Regarding displacement angles, we estimate

uncertainties of scene-pair data to be lower than 10° for Sentinel-1 12-day repeat velocities faster than ~0.5 m d[-1] and for Landsat-8 16-day repeat velocities faster than ~1 m d[-1].

## 3.3 Comparison of Sentinel-1 and Landsat-8 yearly velocity mosaics

Figure 8 shows the example of a 2019 Sentinel-1 velocity mosaic over Svalbard and a selection of statistical measurements that are regularly provided along with the main velocity products. Except of a very small area over Austfonna Basin 3 (~6

km$^2$), the mosaic provides full coverage of velocity information. The mosaic was generated from 557 scene-pair Sentinel-1 velocity fields with a time stamp between 1 January 2019 and 29 December 2019, following the approach described in Sect. 2.3. However, the effective number of measurements per pixel varies regionally (Fig. 8b). This is a) because of different scene availability along different satellite paths and b) because of low image correlation in regions either characterized by surface weathering, snow accumulation and featureless surfaces, such as the interiors of ice caps and the accumulation zones or in

regions where mean flow velocities are very high, such as Austfonna Basin 3.

Standard deviations are ~0.02–0.04 m d[-1] in x-direction and ~0.04–0.08 m d[-1] in y-direction (~0.04 m d[-1]–0.09 m d[-1] for the velocity magnitude) over stable ground and in very slow moving areas for a mosaic's average time separation of ~8 days. The standard deviation differences between both directions reflect the differences in accuracy caused by the different azimuth and range resolutions of the data. The values correspond well to the average statistical velocity magnitude measures that others

(Strozzi et al., 2017; 0.05–0.08 m d-1) and we (Sect. 3.2; ~0.04 m d[-1]) derived for scene-pair velocity fields with a time separation of 12 days over such areas on Svalbard. However, on the glacier tongues, standard deviations sometimes exceed 0.4 m d[-1] in both directions, which is mainly due to the strong intra-annual velocity variations of most of the glaciers (Sect. 3.1). As the standard error is dependent on both the standard deviation and the amount of measurements, it is generally larger for measurements in the y-direction and in regions of few measurements. Nevertheless, while allowing for formal propagation

of errors, standard errors are typically unrealistically low and underestimate the real error in velocity, especially in case of a large number of measurements. However, using standard errors along with the measurement count provides a good qualitative metric for identifying areas of poor measurements.

To assess the difference between Sentinel-1 and Landsat-8 annual velocity mosaics, we compare our weighted mean of the 2018 velocity magnitude (Figure 9a) with that derived from Landsat-8 ITS_LIVE for the same year (Figure 9b). Mean and

median differences (Sentinel-1 minus Landsat-8) are -0.0039 and -0.0004 m d[-1], with a standard deviation of 0.1247 m d [-1] and a NMAD of 0.0143 m d[-1], respectively (Figure 9e). Overall, we find that both data sets are in good agreement. Absolute velocity differences are generally less than 0.02 m d[-1] over stable ground and slow moving ice with enough surface features that can be successfully tracked. However, in the very slow moving accumulation areas of some ice fields and ice caps, Landsat-8 velocities are up to more than 0.1 m d[-1] higher than those of Sentinel-1 (Figure 9c). While radar speckle tracking



derives useful results here, the Landsat-8 mosaic has considerable blunders (Figure 9b), as these regions are difficult for optical feature tracking due to frequent cloud coverage and low-feature surfaces. Additionally, we find absolute differences of >0.2 m d$^{-1}$ over glaciers that have considerable seasonal velocity variations or a surging behavior. Here, several factors take effect: While we calculated the mean surface velocity by SNR weighting, the ITS_LIVE mosaic was derived by error weighting. Hence, in case of the ITS-LIVE mosaic, velocity fields with largest temporal baselines (i.e. smallest theoretical errors) and

consequently heavily temporally smoothed velocities, have the biggest influence on the overall mean. In general, the time separation of the Landsat-8 input image pairs is much larger (16 – 546 days). In contrast, as one of the main focus of our data set is to provide glacier velocity time series at very high temporal resolution, much more velocity fields with considerably shorter temporal baselines (mostly 6 – 12 days) went into the mean calculations of the Sentinel-1 mosaics, leading to a bigger influence of short term velocity variations on the overall mean. Additionally, since Landsat-8 has no coverage in Polar Regions

during wintertime, a general bias towards summertime velocities is expected for the ITS_LIVE data. As acceleration and velocity peaks during spring and summer are typical seasonal glacier velocity signals in Svalbard (Sect. 3.1 and Fig. 5), the Landsat-8 mean velocities tend to be higher than those of Sentinel-1 on some glaciers. The same applies to some surging glaciers, where the surge velocity signal is overlain by summer acceleration peaks. Additionally, on some glaciers in their post surge phase, like e.g. Tunabreen, rapid deceleration took place in autumn 2018, followed by very low winter velocities that

are captured by many single measurements in our data set (Fig. 5e). Furthermore, different spatial resolutions of the mosaics and different processing parameters (e.g. window sizes) lead to some velocity differences in regions where pixels contain mixed information of high and (very) low velocities, like e.g. shear margins.

Additionally, we compare displacement angles relative to true north as derived from the 2018 x- and y-velocity mosaics of both data sets. Since displacement angles are generally unreliable in regions of very slow ice flow (Sect. 3.2), we confined our

analysis to pixels with a mean velocity magnitude of >0.3 m d$^{-1}$ in the RETREAT mosaic (Fig. 9d). We find mean and median differences of -0.67° and -0.94°, with a standard deviation of 8.86° and a NMAD of 5.52°, respectively (Fig. 9f). We therefore conclude that despite of some differences in the mean velocity magnitude on some glaciers mostly due to large inter-annual velocity variations, displacement angles of both mosaics are in very good agreement.

**4 Conclusions and Outlook**

We presented a new data set of scene-pair, as well as monthly and annually averaged 200 m ice velocity grids. We derived the velocity information by applying intensity feature and speckle tracking to all available Sentinel-1 radar images over 12 glaciated regions outside the large polar ice sheets. Our data spans the period from 2014–today and is continuously updated as soon as new data is available. By making all data freely accessible via our interactive web portal, our work is a valuable contribution to open science.

In contrast to existing data sets based on Landsat imagery, we are able to provide continuous glacier velocity-time series all year round independently from weather conditions and sun illumination, at very high temporal resolutions of up to <6 days in



regions where 6-day repeat data is available for overlapping orbits. Using the example of Svalbard, we demonstrated that our dense velocity time series are able to capture seasonal velocity fluctuations, as well as surges and long-term velocity trends at unprecedented temporal detail. This makes our data set particularly suited for detailed investigations and continuous

monitoring of short-term glacier dynamics (e.g. surges, changes in seasonal flow regimes) and long-term velocity trends, as well as their associated drivers. We also see great potential for combining our dense velocity time series with methods from the emerging field of artificial intelligence, e.g. to implement an early warning system for regions of surging glaciers. A comparison of our 12-day repeat Sentinel-1velocities with those generated from very high-resolution 11-day repeat TerraSAR-X data revealed an empirical mid-glacier velocity uncertainty of <0.08 m d$^{-1}$ that is lower than the theoretical uncertainty

(~0.12 m d$^{-1}$) and less than half of the uncertainty that we determined for velocities derived from 16-day repeat Landsat-8 data (0.17–0.18 m d$^{-1}$). Off-glacier velocity differences between Sentinel-1 and TerraSAR-X data of <0.04 m d$^{-1}$ are even 5–6 times lower than those measured for Landsat-8 velocity fields (~0.19–~0.24 m d$^{-1}$). Overall, we find that our Sentinel-1 scene-pair velocities are an excellent complement to the already existing Landsat-8 scene-pair velocity data sets.

Furthermore, our Sentinel-1 velocity mosaics provide smooth and nearly complete velocity information throughout the glacier

areas at annual and monthly resolution. It offers wide application in numerical ice dynamic modeling and mass flux calculations. They complement well with mosaics derived from Landsat-8 data, since we see an advantage over featureless and slow moving ice caps interiors and accumulation areas, where speckle tracking on Sentinel-1 6-day repeat acquisitions derives more reliable velocity measurements than the optical data.

In the future, the data set may be extended by more precise velocity measurements in very slow moving regions, derived by

applying DInSAR (Differential Interferometric SAR) techniques. Furthermore, data collected by previously operating radar satellites (e.g. ERS-1/2, 1991–2011), as well as new (e.g. RADARSAT Constellation, since 2019) and upcoming missions, like the joint NASA-ISRO (Indian Space Research Organisation) SAR mission (NISAR) can be integrated into our processing chain. This would further increase the temporal resolution of our velocity data and its temporal coverage.

**Data availability**

Free access to the complete global Sentinel-1 velocity data set is provided via an interactive web portal (http://retreat.geographie.uni-erlangen.de) after user registration. During the review process, the subset of Svalbard analyzed in this paper is additionally available at the GFZ Potsdam Data Services via a temporary link (https://dataservices.gfz-potsdam.de/panmetaworks/review/18399225c3952e603d6c31555ca146b5458e566074d5d1dbeb2dbcbdca8d0623/). Once the

manuscript is accepted, the Svalbard data set will be published under the DOI https://doi.org/10.5880/fidgeo.2021.016 (Friedl et al. 2021). The raw Sentinel-1 IW SLC acquisitions are available at the ASF DAAC (https://search.asf.alaska.edu, last access: 29 March 2021).

TerraSAR-X SM acquisitions are available via the DLR EOWEB Geoportal (https://eoweb.dlr.de/egp/, last access: 29 March 2021) after submission of a scientific proposal at the TerraSAR-X science service system (https://sss.terrasar-x.dlr.de/, last

access: 29 March 2021). The global TanDEM-X DEM at 3 arc second resolution is available at



https://download.geoservice.dlr.de/TDM90/ (last access: 29 March 2021). The the void-filled 3 arc second global NASA SRTM DEM Version 3 is available via the NASA EARTHDATA portal (https://search.earthdata.nasa.gov, last access: 29 March 2021). The RGI 6.0 data sate is available at https://www.glims.org/RGI/ (last access: 29 March 2021). The HydroLAKES data set can be downloaded at https://www.hydrosheds.org/page/hydrolakes (last access: 29 March 2021). The

OpenStreetMap-based land- and ocean masks are available at https://osmdata.openstreetmap.de/data/land-polygons.html and https://osmdata.openstreetmap.de/data/water-polygons.html (last access: 29 March 2021), respectively. The Svalbard glacier centerlines were provided upon request by the authors of Nuth et al. (2013). The ITS_LIVE and GoLIVE Landsat-8 ice surface velocity products are available at https://nsidc.org/apps/itslive/ and http://nsidc.org/app/golive, respectively.

**Author contributions**

PF was the principal investigator of the project, conducted the analyses, developed the Senintel-1 processing chain on the HPC, produced all data and figures and wrote the manuscript. TS contributed to the generation of the displacement angle maps and the writing of the manuscript. MB had the project idea, organized the funding and contributed to the writing of the manuscript. All authors revised the manuscript.


**Acknowledgements**

We acknowledge the kind provision of the radar satellite data via the freely accessible ASF DAAC (Sentinel-1) and the DLR proposal HYD1763 (TerraSAR-X). We thank NASA for making the GoLIVE and ITS_LIVE Landsat-8 ice surface velocity and the SRTM-DEM data freely available, as well as DLR for providing the global TanDEM-X DEM free of charge.

Furthermore, we thank the authors of the HydroLAKES data set, the Svalbard glacier centerlines, the RGI 6.0 and FOSSGIS e.V. (https://github.com/fossgis; https://osmdata.openstreetmap.de) for sharing their data.

The authors would like to thank DLR/BMWi for funding this activity under the project RETREAT (FKZ 50EE1716).

**Competing interests**

The authors declare that they have no conflict of interest.

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

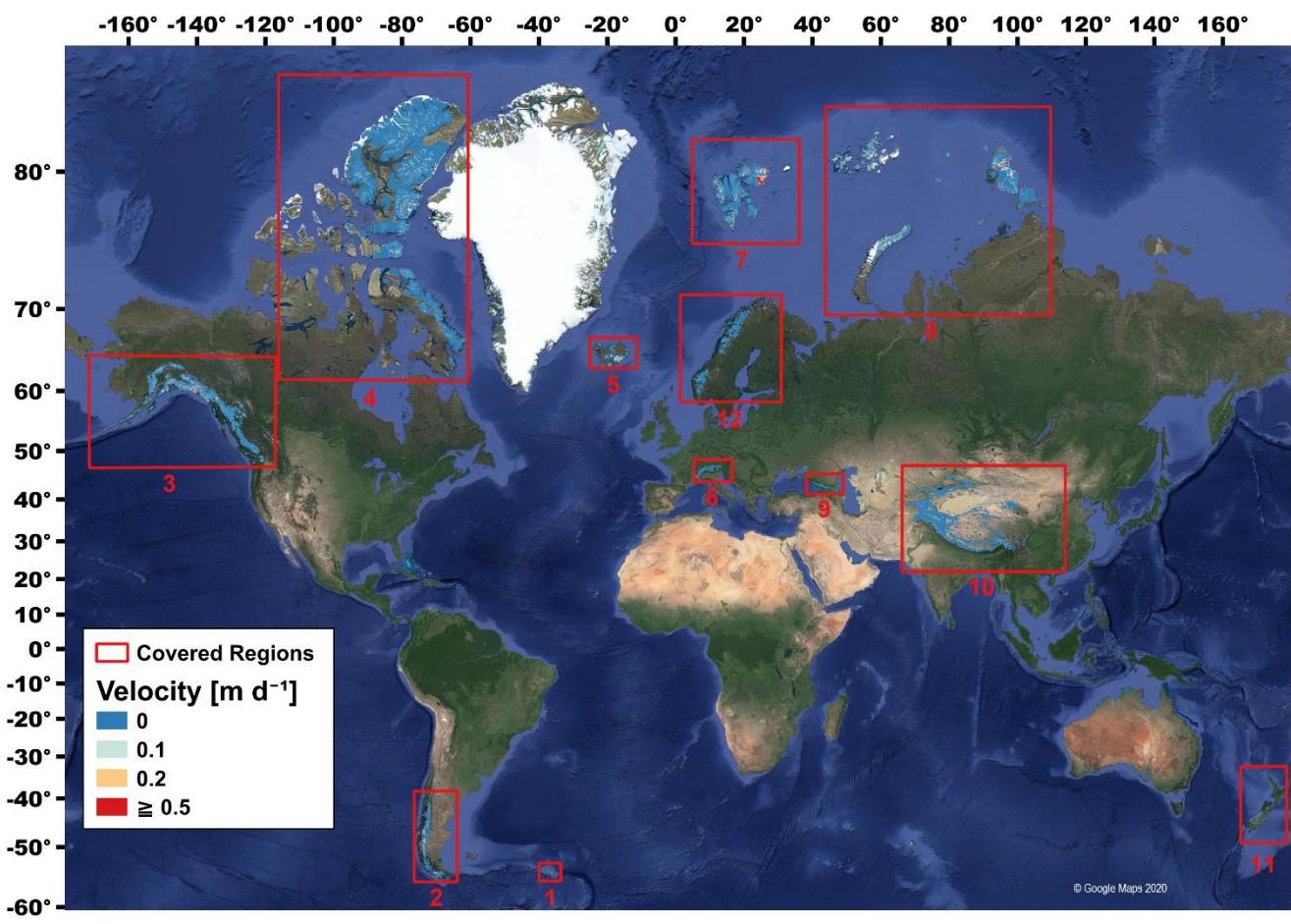

**Figure 1: 12 Regions covered by the RETREAT data set.** Velocities are shown as annual Sentinel-1 velocity mosaics of 2019. Map base: © Google Maps 2020.



**Figure 2: Examples of raw, filtered and corrected single velocity fields.** Raw velocity (magnitude) fields from 15 January 2019 (6-day repeat pair, **a₁**) and 11 August 2019 (12-day repeat pair, **b₁**). Filtered products using the approach of Lüttig et al. (2017) are shown in panels **a₂** and **b₂**. The products that were corrected for coregistration errors are displayed in panels **a₃** and **b₃**. The short 6-day temporal baseline of the 15 January 2019 velocity field and the absence of surface melt during winter allowed for tracking radar speckle on the almost featureless surface of Vestfonna ice cap. Filtering effectively removed existing blunders while keeping reliable measurements in both examples. While coregistration error correction significantly improved the velocity field on Vestfonna **(a₃)**, it had only minor effects on the velocities from 11 August 2019 **(b₃)** due to abundant ice free area in this scene. The coregistration error correction values applied to both velocity fields are shown in Fig. 3. Background: © Google Maps 2020.




**Figure 3: Determination of coregistration error correction values.** Coregistration error correction values in az and r direction are shown for the velocity fields from 15 January 2019 (($a_1$) and ($a_2$)) and 11 August 2019 (($b_1$) and ($b_2$)) displayed in Fig. 2. Black circles represent displacement values extracted over stable ground, plotted against their corresponding SNR (Signal to Noise Ratio) values, computed as SNR=CCP/CCS. The median displacement values (red lines) coincide well with the displacement values that have the highest SNR values
(i.e. the most reliable measurements, assumed to be primarily affected by coregistration inaccuracy). After correction, median displacement values are 0 for both directions.

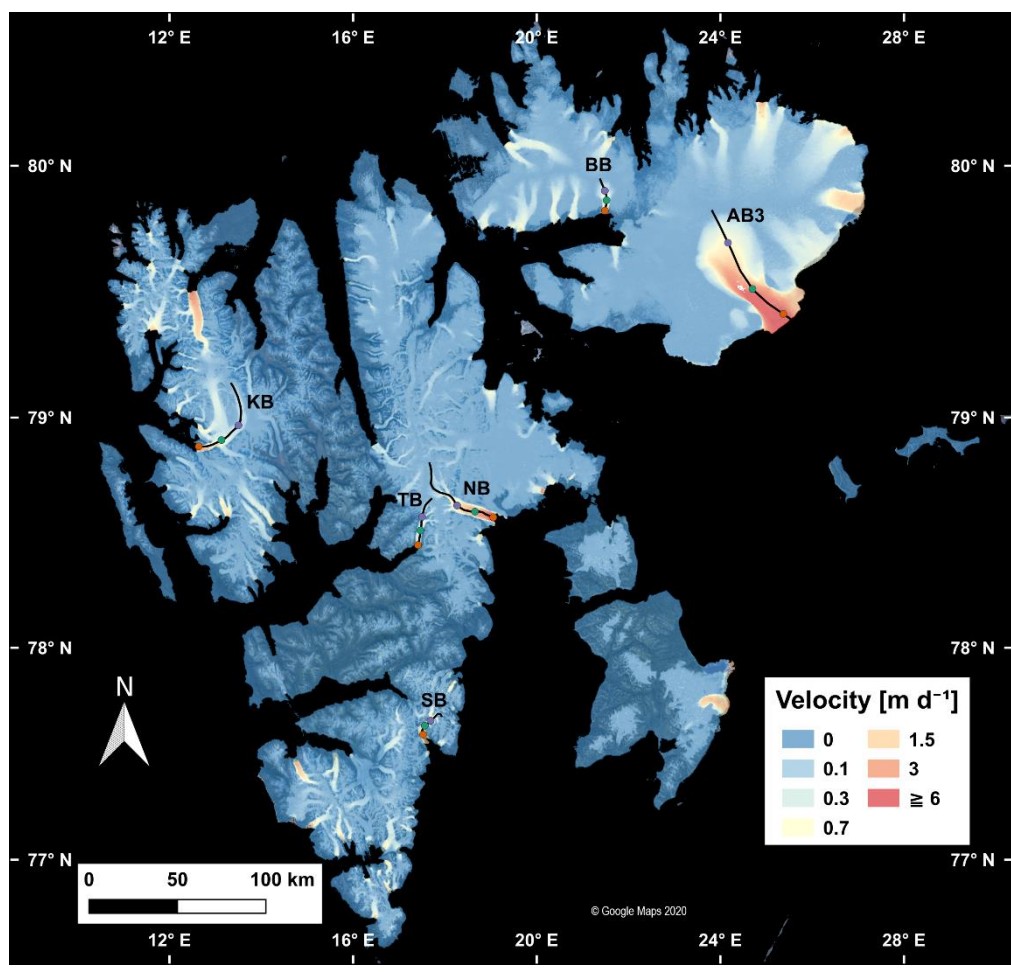

**Figure 4: Overview map of Svalbard showing the locations of velocity extraction points on a Sentinel-1 velocity mosaic from 2019.**
Surface velocities were extracted for 6 glaciers: Austfonna Basin 3 (AB3), Bodleybreen (BB), Kronebreen (KB), Negribreen (NB), Tunabreen (TB) and Strongbreen (SB). The color coding of the extraction points, distributed along the glacier's centerlines, is the same as for the corresponding velocity time series in Fig. 4. Background: © Google Maps 2020, overlain by an OpenStreetMap-based ocean polygon data set in black (https://osmdata.openstreetmap.de/data/water-polygons.html, © OpenStreetMap contributors 2019. Distributed under the Open Data Commons Open Database License (ODbL) v1.0).







**Figure 5: Velocity time series from Sentinel-1 (1 January 2015–30 November 2020) for 6 different glaciers in Svalbard.** The color coding is the same as for the corresponding extraction points in Fig. 4.





**Figure 6: Comparison of scene-pair velocity fields from Sentinel-1, Landsat-8 ITS_LIVE and Landsat-8 GoLIVE with TerraSAR-X over 4 different glaciers in Svalbard.** TSX: TerraSAR-X, S1: Sentinel-1, L8 IL: Landsat-8 ITS_LIVE, L8 GOL: Landsat-8 GoLIVE. The dates of acquisition are listed in Table 1. Background: © Google Maps 2020


**Figure 7: Comparison of velocity and displacement angle profiles from TerraSAR-X, Sentinel-1, Landsat-8 ITS_LIVE and Landsat-8 GoLIVE data sets, extracted along the centerlines of 4 different glaciers in Svalbard.** Centerlines are displayed in Fig. 6 and the dates of acquisition are listed in Table 1.






**Figure 8: Sentinel-1 surface velocity mosaic generated from 557 scene-pair velocity fields with a 2019 time stamp. a)** velocity
magnitude, **b)** measurement count per pixel, **c)** weighted standard deviation of the x velocity component, **d)** weighted standard deviation of
the y velocity component, **e)** weighted standard error of the x velocity component, **f)** weighted standard error of the y velocity component.
Background: © Google Maps 2020, overlain by an OpenStreetMap-based ocean polygon data set in black

(https://osmdata.openstreetmap.de/data/water-polygons.html,© OpenStreetMap contributors 2019. Distributed under the Open Data
Commons Open Database License (ODbL) v1.0).

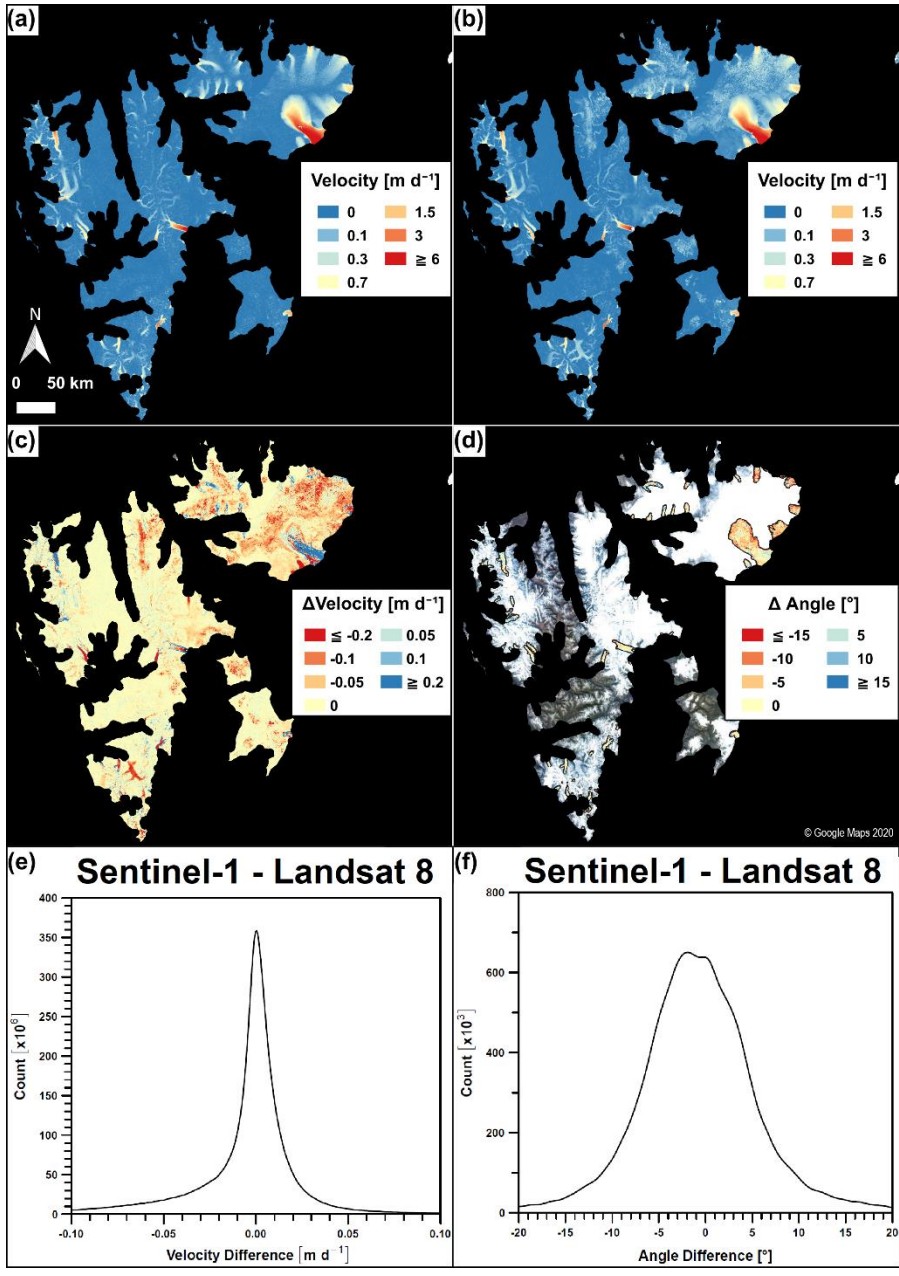

**Figure 9: Comparison of a Sentinel-1 surface velocity mosaic with a Landsat-8 ITS_LIVE velocity mosaic from 2018. a)** Sentinel-1
weighted mean of the velocity magnitude, **b)** Landsat-8 weighted mean of the velocity magnitude, **c)** velocity magnitude difference
(Sentinel-1 minus Landsat-8) **d)** displacement angle difference in regions with surface velocities of $> 0.3$ m d$^{-1}$(Sentinel-1 minus Landsat-
8) **e)** distribution of the velocity differences, **f)** distribution of the displacement angle differences. Background: © Google Maps 2020,
overlain by an OpenStreetMap-based ocean polygon data set in black (https://osmdata.openstreetmap.de/data/water-polygons.html,©
OpenStreetMap contributors 2019. Distributed under the Open Data Commons Open Database License (ODbL) v1.0).





**790**     **Table 1: Results of the intercomparison experiment of Sentinel-1, Landsat-8 ITS_LIVE and Landsat-8 GoLIVE ice velocity and displacement angle fields with TerraSAR-X.** The corresponding extraction areas are shown in Fig. 6. Minimum differences to TerraSAR-X are highlighted as bold text.

| (a) Kronebreen | | | | | | | |
|---|---|---|---|---|---|---|---|
| **Dataset** | **Acquisition dates [yyyy-mm-dd]** | **Front/shear zones median difference** | **Front/shear zones NMAD** | **Mid-glacier median difference** | **Mid-glacier NMAD** | **Stable ground median difference** | **Stable ground NMAD** |
| TerraSAR-X | 2018-03-06 2018-03-17 | - | - | - | - | - | - |
| Sentinel-1 | 2018-03-07 2018-03-19 | -0.067 m d⁻¹ -3.81 ° | 0.222 m d⁻¹ 5.74 ° | **0.005 m d⁻¹** **-2.42 °** | **0.077 m d⁻¹** **4.71 °** | **-0.049 m d⁻¹** | **0.049 m d⁻¹** |
| Landsat-8 ITS_LIVE | 2018-03-09 2018-03-25 | 0.090 m d⁻¹ -10.87 ° | 0.225 m d⁻¹ **3.64 °** | 0.179 m d⁻¹ -23.26 ° | 0.095 m d⁻¹ 6.21 ° | -0.220 m d⁻¹ | 0.066 m d⁻¹ |
| Landsat 8 GoLIVE | 2018-03-09 2018-03-25 | **-0.047 m d⁻¹** **0.02 °** | **0.165 m d⁻¹** 5.04 ° | 0.012 m d⁻¹ -8.19 ° | 0.165 m d⁻¹ 9.37 ° | -0.118 m d⁻¹ | 0.106 m d⁻¹ |

| (b) Negribreen | | | | | | | |
|---|---|---|---|---|---|---|---|
| **Dataset** | **Acquisition dates [yyyy-mm-dd]** | **Front/shear zones median difference** | **Front/shear zones NMAD** | **Mid-glacier median difference** | **Mid-glacier NMAD** | **Stable ground median difference** | **Stable ground NMAD** |
| TerraSAR-X | 2015-02-01 2015-02-12 | - | - | - | - | - | - |
| Sentinel-1 | 2015-02-03 2015-02-15 | -0.028 m d⁻¹ (19.45 °) | 0.088 m d⁻¹ (26.94 °) | -0.015 m d⁻¹ (15.95 °) | 0.072 m d⁻¹ (29.90 °) | -0.064 m d⁻¹ | 0.056 m d⁻¹ |

| (c) Tunabreen | | | | | | | |
|---|---|---|---|---|---|---|---|
| **Dataset** | **Acquisition dates [yyyy-mm-dd]** | **Front/shear zones median difference** | **Front/shear zones NMAD** | **Mid-glacier median difference** | **Mid-glacier NMAD** | **Stable ground median difference** | **Stable ground NMAD** |
| TerraSAR-X | 2017-03-27 2017-04-07 | - | - | - | - | - | - |
| Sentinel-1 | 2017-03-30 2017-04-11 | **0.061 m d⁻¹** **-0.54 °** | **0.262 m d⁻¹** 8.00 ° | **-0.015 m d⁻¹** **-1.70 °** | **0.098 m d⁻¹** 14.11 ° | **-0.027 m d⁻¹** | **0.019 m d⁻¹** |
| Landsat-8 ITS_LIVE | 2017-03-21 2017-04-06 | 0.259 m d⁻¹ -1.09 ° | 0.377 m d⁻¹ **4.50 °** | 0.096 m d⁻¹ 34.68 ° | 0.295 m d⁻¹ 127.10 ° | -0.193 m d⁻¹ | 0.083 m d⁻¹ |
| Landsat 8 GoLIVE | 2017-03-21 2017-04-06 | 0.075 m d⁻¹ -11.83 ° | 0.343 m d⁻¹ 6.18 ° | 0.159 m d⁻¹ -47.18 ° | 0.189 m d⁻¹ 37.80° | -0.264 m d⁻¹ | 0.124 m d⁻¹ |

| (d) Strongbreen | | | | | | | |
|---|---|---|---|---|---|---|---|
| **Dataset** | **Acquisition dates [yyyy-mm-dd]** | **Front/shear zones median difference** | **Front/shear zones NMAD** | **Mid-glacier median difference** | **Mid-glacier NMAD** | **Stable ground median difference** | **Stable ground NMAD** |
| TerraSAR-X | 2015-10-10 2015-10-21 | - | - | - | - | - | - |
| Sentinel-1 | 2015-10-01 2015-10-13 | **0.015 m d⁻¹** (0.19 °) | **0.04 m d⁻¹** (128.9 °) | 0.036 m d⁻¹ -3.00 ° | 0.072 m d⁻¹ 7.15 ° | **-0.006 m d⁻¹** | **0.019 m d⁻¹** |
| Landsat-8 ITS_LIVE | 2015-09-26 2015-10-12 | -0.335 m d⁻¹ (-56.42 °) | 0.240 m d⁻¹ (71.68 °) | -0.234 m d⁻¹ **-2.64 °** | 0.155 m d⁻¹ **3.58 °** | -0.303 m d⁻¹ | 0.087 m d⁻¹ |
| Landsat 8 GoLIVE | 2015-09-26 2015-10-12 | (-0.314 m d⁻¹) (-93.33 °) | (0.219 m d⁻¹) (94.67 °) | (-0.196 m d⁻¹) (4.68 °) | (0.367 m d⁻¹) (13.99 °) | (-0.380 m d⁻¹) | (0.124 m d⁻¹) |

| Overall average | | | | | | | |
|---|---|---|---|---|---|---|---|
| **Dataset** | | **Front/shear zones median difference** | **Front/shear zones NMAD** | **Mid-glacier median difference** | **Mid-glacier NMAD** | **Stable ground median difference** | **Stable ground NMAD** |
| Sentinel-1 | | **-0.005 m d⁻¹** **-2.18 °** | **0.153 m d⁻¹** 6.87 ° | **0.003 m d⁻¹** **-2.37 °** | **0.079 m d⁻¹** **8.66 °** | **-0.037 m d⁻¹** | **0.036 m d⁻¹** |
| Landsat-8 ITS_LIVE | | **0.005 m d⁻¹** -5.98 ° | 0.281 m d⁻¹ **4.07 °** | 0.014 m d⁻¹ 2.93 ° | 0.182 m d⁻¹ 45.63 ° | -0.239 m d⁻¹ | 0.08 m d⁻¹ |
| Landsat 8 GoLIVE | | 0.014 m d⁻¹ -5.91 ° | 0.254 m d⁻¹ 5.61 ° | 0.086 m d⁻¹ -27.69° | 0.177 m d⁻¹ 23.59 ° | -0.191 m d⁻¹ | 0.115 m d⁻¹ |