# Peer review of "Global time series and temporal mosaics of glacier surface velocities, derived from Sentinel-1 data"

_Earth System Science Data, 2021_

## Author Comment (AC1)

A new global data set of Sentinel-1 glacier surface velocities that covers 12 major glaciated regions outside the polar ice sheets is presented. This is a very welcomed data set, congratulations to the authors for their efforts! By making all data freely accessible via the interactive web portal, your work is indeed a valuable contribution to open science. The portal with the Sentinel-1 glacier surface velocities is intuitive and data sets can be easily searched, downloaded and analysed. A comparison with velocity maps we produced within the ESA Glaciers_CCI project (https://climate.esa.int/en/projects/glaciers) revealed an outstanding quality of the data.

The manuscript is carefully written with very accurate descriptions of methods and results. I have three major comments and a series of small amendments and suggestions to be included in a minor revision of the manuscript.

1. Agreed that with your data set you are able to provide continuous glacier velocity-time series all year round independently from weather conditions and sun illumination, but the quality of results obtained from Sentinel-1 is better in winter than in summer. In particular, often during the summer the coverage with valid data is more restricted than in winter because of snow and ice melt and thus loss of coherence (or speckle). This point should be somewhere mentioned in the description of the data.

**We agree with the reviewer. We added the following sentence to the Data and Methods section:** *In general, the quality of tracking results is often better in winter than in summer over both accumulation areas and glacier tongues, because snow and ice melt during summer can quickly alter the surface properties of tracking features (i.e. feature tracking becomes more difficult) and cause loss of coherence (i.e. speckle tracking becomes infeasible).*

2. What is the effect of the coarse Sentinel-1 spatial resolution on small glaciers, in particular over mountainous areas? Is there a minimum size below which the Sentinel-1 results are no more accurate? Is there an underestimation of the ice velocity measured with Sentinel-1 over small glaciers? Please discuss this point in your manuscript. One way to study the performance of Sentinel-1 over small mountainous glaciers would be a comparison to Sentinel-2, e.g. over fast moving surging glaciers in the Himalayas.

**We appreciate the reviewers comment. Following the suggestion by the reviewer, we carried out a comparison of Sentinel-1 results with other data at a mountain glacier. We selected Yazghil Glacier in the Karakoram Range. Compared to the many wide tide water glaciers in Svalbard, is Yazghil Glacer a rather narrow (~800 m) but fast flowing (up to 2.5 m/d) mountain glacier, allowing an analysis of the impact of the spatial resolution and the tracking window size of our products for smaller glaciers.**

**However, we decided to do a comparison with velocities derived from high resolution TerraSAR-X/TanDEM-X (TSX in the following) data (~3 m ground range resolution). Results, the discussion and conclusion are provided below.**

**We added the following statement to the manuscript.**

**Supplement:**

*Section S1: Comparison between Sentinel-1 and TerraSAR-X velocity fields at a narrow mountain glacier.*

*We carried out a comparison of our Sentinel-1 products with a velocity field derived from high resolution TerraSAR-X data at the glacier tongue of Yazghil Glacier in December 2015. Yazghil Glacier is a rather narrow (~800 m) but partly fast flowing (up to 2.5 m $d^{-1}$) mountain glacier in the Karakorum, allowing an analysis of the impact of the spatial resolution and the tracking window size on the velocity estimates.*

*We applied the same routine to the TerraSAR-X data as for the comparison on Svalbard (Section 2.5). The resulting velocity fields from both sensors are illustrated in Figure S1. We selected 4 profiles (one along and three cross glacier profiles) to further investigate the difference between the resulting products (Figure S2). As seen in Figure S1, the Sentinel-1 product has a lower spatial coverage on glacier. Moreover, the velocities in the lateral zones of the glaciers are often lower for Sentinel-1 as for TerraSAR-X. These findings are supported by the velocity measurements along the glacier profiles (Figure S2). In particular the plot of profile 1 indicates that there is a general trend for Sentinel-1 products to underestimate the flow velocity, in particular at the slow-moving section towards the terminus and for the small-scale high velocity section.*

*The larger tracking window size for Sentinel-1 (250 x 50 pixels at about 4x20 m ground resolution, see Table S1 vs. 128x128 pixels at about 3x3 m ground resolution for TerraSAR-X), explains these limitations. For Sentinel-1, more stable areas around the glacier are covered by the tracking window, affecting the offset estimation and leading to the revealed but also expected underestimation of the glacier velocities towards the margins. In a similar way, the relatively small zone with high glacier velocities (>1 m $d^{-1}$ according to the TerraSAR-X data) are partly averaged out by the lower resolution and larger tracking window size of the Sentinel-1 data.*

*The tracking window sizes were carefully selected to obtain good results for the majority of the glaciers in the respective region. However, for such a global database, there are always certain limitations and not all details can be fully resolved. Considering the results from this comparison and experience from visible inspection of further Sentinel-1 products, we conclude that it is very likely, that the flow velocities at glaciers narrower 1 km are underestimated, in particular towards the margins, and that high velocity variations at scales of 1-2 km and below may be partly averaged out.*

[Figure]

*Figure S1: Surface velocity fields at Yazghil Glacier, Karakorum, derived from Sentinel-1 (left panel) and TerraSAR-X (right panel) imagery in December 2015. Black lines indicate profiles for velocity measurements, see Figure S2. Background: Bing Satellite*

[Figure]

*Figure S2: Velocity estimates based on Sentinel-1 (blue dots) and TerraSAR-X (red dots) along profiles on Yazghil Glacier (See Figure S1) in December 2015. Vertical lines in the top right panel indicate crossing points with other profiles.*

**Manuscript:**

**End of Section 3.2**

*In order to investigate the impact of the tracking window size and spatial resolution of the Sentinel-1 data on the quality of the results for narrow glaciers, we carried out a comparison between Sentinel-1 and TerraSAR-X velocity fields at the glacier tongue of Yazghil Glacier in the Karakorum. We conclude that it is very likely for glaciers narrower 1 km, that the velocity estimates are underestimated, in particular towards the margins, and that small (< 1-2 km) velocity fluctuations may be partly averaged out. We attribute both issues to the lower spatial resolution of the Sentinel-1 acquisition in combination with the used tracking window size. More details on the analysis can be found in Supplement Section S1.*

3. I couldn't find any indication in your paper about use of ascending and descending Sentinel-1 data. Are you processing both directions and combine the results? Or just of the two? Have you done any comparison, e.g. over Svalbard? And for other regions? In principle, ascending and descending Sentinel-1 data could be also employed for a 3D decomposition of the ice velocity vector.

**At the moment, we do not combine ascending and descending data but use consecutive pairs of images with the same imaging geometry. However, we agree with the reviewer that combining ascending and descending passes is a promising technique to further increase the quality of the measurements and hence an interesting feature to be included into our processing chain in the future. We changed the sentence in l. 73 ff. to:** *Our main input data are consecutive pairs of single or dual polarized Sentinel-1 SLC (Single Look Complex) SAR (Synthetic Aperture Radar) images with the same imaging geometry, acquired over 12 glacierized regions outside the Antarctic and Greenland ice sheets (Fig. 1). Ascending and descending orbits are handled independently.*

**We also added a sentence to the Conclusions and Outlook section:** *In the future, the data set may be extended by more precise velocity measurements, derived by applying DInSAR (Differential Interferometric SAR) techniques in very slow moving regions and by combining acquisitions from ascending and descending satellite passes (Sánchez-Gámez and Navarro, 2017).*

Here the list if minor points:

l. 11. What do you mean by "near" global? What is not covered? Certain regions? Or glaciers, e.g. small ones?

**The data set is "near" global, since the polar ice sheets and surrounding glaciers and ice caps are not included. However, we removed the word "near", as the spatial limitations of the data set are explained in l. 14 ff.**

ll. 13-14. By writing that velocity is derived "by applying feature and speckle tracking" you give the impression that two algorithms are used. Instead, the technique is based on tracking persistent patterns of intensity values in both images, which are either formed by surface features such as crevasses (feature tracking) or correlated radar speckle (speckle tracking). This point can be better explained also in the abstract. Same applies to the conclusions (l. 481).

**We changed the sentence in the abstract to:** *The velocity information is derived from archived and new Sentinel-1 SAR acquisitions by applying a well-established intensity offset tracking technique.* **In the conclusions we now write:** *We derived the velocity information by applying intensity offset tracking to all available Sentinel-1 radar images over 12 glacierized regions outside the large polar ice sheets.* **This leaves the exact tracking procedure somewhat open, but does not suggest anymore that two separate algorithms are used. The detailed explanation of what is tracked by the algorithm (either surface features or speckle) is contained in the text (L 121 ff.).**

ll. 38-39. Actually, Ice Velocity is a product of the ECV Ice Sheets and Ice Shelves (https://gcos.wmo.int/en/essential-climate-variables/ice-sheets-ice-shelves). For glaciers, the products are only Glacier Area, Glacier Elevation Change and Glacier Mass Change (https://gcos.wmo.int/en/essential-climate-variables/glaciers/ecv-requirements).

**We thank the reviewer for this hint. We changed the corresponding sentence to:** *Therefore, glacier surface velocity and its short and long-term variations should be monitored on a regular and global scale.*

l. 30. Why "only"? I would remove this adverb.

**We think that the reviewer refers to l. 40 instead of l. 30. We removed the word "only" here.**

l. 48. In addition to sun illumination and polar night, past coverage of optical velocity data is restricted by other constraints of historical missions (e.g. acquisition capacity, image quality, …).

**We thank the reviewer for this important note. We added the following sentence to the text:** *Furthermore, past coverage of optical velocity data is restricted by the general constraints of historical satellite missions such as e.g. acquisition capacity and image quality.*

l. 83. Add "slant" to the resolution of about 3 m.

**Added.**

l. 84. You can possibly explain why data are available at HH or VV polarization, i.e. VV polarization is the default mode over land, while HH polarization is the default mode over polar regions (see e.g. https://sentinels.copernicus.eu/web/sentinel/missions/sentinel-1/observation-scenario).

**We agree with the reviewer that the coverage of the different polarization modes is worth to mention. We added the following sentence to the text:** *The HH or HH-HV polarization is the standard polarization scheme for acquisitions over polar regions and the VV or VV-VH polarization is the default mode for all other observation zones.*

ll. 205-207. If you want, you can add here that by applying this procedure the possible bias introduced by strong, short-term summer speed-up events is removed from the annual means.

**Good point. We added this sentence:** *This procedure removes the possible bias introduced by strong, short-term summer speed-up events (Sect. 3.1) from the annual means.*

ll. 285-287. I would not consider the surges of Austfonna Basin 3 and Negribreen too similar, because the stepwise frontal acceleration of Austfonna Basin 3 lasted at least five years with winter velocities much smaller than summer ones, while that of Negribreen lasted only two years with only a slight slow-down during winter.

**We agree with the reviewer. We added the following sentence, in order to more emphasize the differences between the surges of both glaciers:** *In contrast to the surge of Austfonna Basin 3, the stepwise acceleration phase of Negribreen was shorter (2 instead of 5 years) and the difference between summer and winter velocities during the acceleration phase was much more pronounced on Austfonna Basin 3 (Dunse et al., 2015).*

l. 302. Tunabreen already surged in 2004, see https://doi.org/10.1016/j.quascirev.2014.11.006, that may explain the short duration of only 2 years, the relatively low maximum velocities, and the absence of a clear seasonal velocity pattern.

**We thank the reviewer for this important hint. We added the following sentence to the text:** *The special characteristics of the surge of Tunabreen with its short duration, the relatively low maximum velocities and the absence of a clear seasonal velocity pattern may be linked to its short temporal distance to the glacier's last surge in 2004 (Flink et al., 2015).*

l. 384. Sentinel-1 and not Setinel-1.

**Changed.**

l. 487. What do you mean by 6-day repeat data available for overlapping orbits? 6-day repeat is over the same orbit.

**We mean that if the same point on earth is sampled by multiple overlapping orbits with a 6-day repeat cycle, the sampling interval of this point can be reduced to < 6 days due to converging orbit at higher latitudes (i.e. the same point is overflown by the satellite e.g. every 2 days). We changed the sentence, hoping that it makes the point clearer:** *In contrast to existing data sets based on Landsat imagery, we are able to provide continuous glacier velocity-time series all year round independently from weather conditions and sun illumination, at very short sampling intervals of up to <6 days over regions that are covered by multiple overlapping orbits with a 6-day repeat cycle.*

l. 506. Also the archives of past JERS-1 SAR data (1992-1998) are now freely and openly available.

**We have added this to the text:** *Furthermore, data collected by previously operating radar satellites (e.g. ERS-1/2, 1991–2011 or JERS-1 SAR, 1992–1998), as well as new (e.g. RADARSAT Constellation, since 2019) and upcoming missions, like the joint NASA-ISRO (Indian Space Research Organisation) SAR mission (NISAR) can be integrated into our processing chain.*

---

## Author Comment (AC2)

The global time series glacier surface velocities were derived from Sentinel-1 data. Scene-pair velocity, as well as monthly and annually averaged velocity mosaics products at 200 m resolution acquired by this study could be easily accessed in the http://retreat.geographie.unierlangen.de with rich quality parameters. Due to the independent of weather conditions, season and daylight of SAR images, the product acquired by this study was a great supplement with improvements in cloud covered regions relative to ITS_LIVE and GoLIVE using optical images.  And this product would be of great use for studies in response of glaciers to climate change and ice thickness inversion and so on.

We would like to thank the reviewer for for his positive assessment of our manuscript and his helpful comments on our manuscript.
All suggested changes were considered and a list of responses and changes in the manuscript is given below. Responses are written in **bold** face type and changes in the manuscript are written in *blue*.

I have only one mainly comment, glaciers in Svalbard are selected to demonstrate the quality of the studies, and shown large superiority in accuracy and temporal resolutions; but the superiority of SAR images independently from weather conditions were not fully analyzed at present. For example, the mountain glaciers on the South-east Tibet were seriously affect by the cloud, both ITS_LIVE and GoLIVE shown weakness in this area. Improvements in like these issues should be fourthly analyzed.

**We appreciate the reviewer's comment. As stated in the paper, due to the physical properties of radar waves, SAR sensors can work weather and cloud independent and are not limited by (polar) night, thus allowing an all year monitoring in particular in polar regions, but also in frequently cloud covered regions such as high mountain areas. This is a well-known and widely accepted advantage of SAR sensors that has already been investigated and discussed in many publications - especially in basic literature on SAR. We therefore think that it is sufficient to add references to the text for further reading on this topic:**

*In contrast, repeat-pass Synthetic Aperture Radar (SAR) data acquired by the Sentinel-1 constellation enable near real time-like and fully automatic processing of global glacier velocities at up to 6-day temporal resolution, independent of weather conditions, season and daylight (Jawak et al., 2015; Moreira et al., 2013) from 2014 until today.*

**However, we conducted a small experiment that demonstrates the general advantage of SAR data regarding cloud coverage. We compared the query results of Landsat-8 and Sentinel-1 data for the year 2020 over an area in South-East Tibet (see figure below). On ASF-Vertex 227 Sentinel-1 SLC scenes with a repeat cycle of 12 days were found over the region, which all can be used without restrictions for velocity calculations. In contrast, on USGS EarthExplorer, no scenes with zero cloud cover were found for Landsat-8 and only 57 scenes with <= 10% and 71 scenes with <=20% cloud cover, respectively. When comparing the query results, it must be also considered, that the temporal baseline of Landsat-8 repeat images with low cloud cover can be up to some hundred days and that the spatial coverage of each Landsat-8 scene is only 185 x 180 km (250 x 250 for Sentinel-1).**

[Figure]

And I also noted that glaciers in the south-east Tibet shown in the website are only partly covered by the red polygons? why?

**The reviewer is right that some of the smaller glaciers in South-East Tibet seem to be only partly covered by the glacier outlines on our website. The data source of our glacier polygons is the RGI 6.0, which we roughly adjusted to the coverage of our velocity data (i.e. glaciers in the periphery of Greenland are for example not covered by the outlines on the website). However, over South-East Tibet, our polygons have the identical coverage as the RGI 6.0. Hence, any missing or wrong glacier outline in this region is directly attributed to inaccuracies in the RGI 6.0 data set. Still, the RGI 6.0 is the most comprehensive glacier data base that is currently available. Regarding the reasons for the inaccuracies in the RGI 6.0 data set over South-East Tibet, we can only speculate, but as the RGI is based on optical data, cloud coverage and discrimination problems between snow and ice may be part of the story.**

Line 14, "glaciated" means area covered by glacier ice in the past, but not at present. "glacierized" is better.

Cogley, J. G., R. Hock, L. Rasmussen, A. Arendt, A. Bauder, R. Braithwaite, P. Jansson, G. Kaser, M. Möller, and L. Nicholson (2011), Glossary of glacier mass balance and related terms, IHP-VII technical documents in hydrology, 86

**We thank the reviewer for this important remark. We changed this for all occurrences in the manuscript.**

Line 70, A flowchart about your process chains and mosaic of different products would be better

**This is a very welcomed remark. We added a flowchart of our processing chain to the manuscript.**

[Figure]

*Figure 2: Flowchart of scene pair velocity field and temporal velocity mosaic generation*

Line 84, Considering the differences of sensitivity of HH or VV polarizations to glaciers surface with different water contents, any velocity differences detected by these two polarization channels.

**The reviewer is right that the different polarizations have a different sensitivity to the water content of the ice surface. Especially, there may be differences in backscatter and the location of the phase center for different polarizations in the presence of wet snow (but less over bare ice). However, we believe that polarization differences are not a major issue in the case of Sentinel-1 velocities, because:**

1) **There is a general polarization scheme for Sentinel-1 that sets the polarization to HH (or HH-HV) over polar regions and to VV (or VV-VH) for all other observation zones (https://sentinels.copernicus.eu/web/sentinel/missions/sentinel-1/observation-scenario). Hence, there is in general no mixture of HH and VV polarizations over the same region.**
2) **As the polarimetric signature of the scatterer does not change within a region, there is no change in the location of the phase center for consecutive images pairs that may be related to a change in the polarization mode. Forming tracking pairs**

**from two consecutive images that have different polarizations may be an issue, but is prevented by the polarization scheme mentioned above. We thus follow the widely accepted and applied approach of single channel tracking.**

**Furthermore, the polarization scheme prevents the investigation (detection) of possible differences between velocities generated from two consecutive HH-polarized images and those simultaneously generated from two VV-polarized images, which is perhaps the actual answer to the reviewer's question. In response to a comment by Tazio Strozzi, we added the following sentence to the Data and Methods section, which explains the polarization scheme of Sentinel-1:**

*The HH or HH-HV polarization is the standard polarization scheme for acquisitions over polar regions and the VV or VV-VH polarization is the default mode for all other observation zones.*

Line 121, therefore, the velocity of glaciers in the accumulation area would still suffer from problems due to their low contrast?

**This is especially the case during summer, where tracking of the radar speckle is not possible due to decorrelation caused by surface melt and where tracking relies on surface features only. We explain this in the text: "However, since speckle tracking requires phase coherence, its application is often restricted to winter acquisitions when there is no surface melt and to regions where 6 day-repeat data is available and where surface velocities are low (i.e. accumulation areas, ice cap interiors)" In accordance to the suggestions of Tazio Strozzi, we additionally added the following sentence to the data section, which hopefully makes the point mentioned by the reviewer clearer:**

*In general, the quality of tracking results is often better in winter than in summer over both accumulation areas and glacier tongues, because snow and ice melt during summer can quickly alter the surface properties of tracking features (i.e. feature tracking becomes more difficult) and cause loss of coherence (i.e. speckle tracking becomes infeasible).*

Line 140 Is it possible a higher resolution product could be produced, which is of great help for studies focused on local scale.

**In principle, velocity products of higher spatial resolution can be produced. However, it has to be considered that increasing the spatial resolution of the velocity product does not necessarily increase the amount of valid information contained in the data. This is because tracking window sizes in the range of tens to some hundred pixels (i.e. ~ 700 m for Svalbard) and step sizes of around 200 m (50 x 10 pixels) limit the true spatial resolution of the generated velocity data. Additionally, from a computational perspective, raster data of higher resolution have exponentially larger file sizes (which has negative effects on storage capacities and download speeds) and demand exponentially more processing resources, such as RAM and CPU runtime. The demand of computational resources is an important factor, when talking about processing very large global data sets with big data input, but may be less important if processing a few images on local scale. Hence, the 200 m resolution of the velocity data is a compromise resulting from considering all points mentioned above. This compromise must be made by any provider of such large global data sets. However, the spatial resolution of our products is still slightly higher than that of other large velocity data sets that are currently available (240 m – 500 m). There is also the possibility to extend our data set by velocities of higher spatial resolution for small selected regions of special interest or**

**upon request by users. We added a sentence on this to the Conclusions and Outlook section:**

*Velocity data of higher spatial resolution (e.g. 100 m) may be additionally produced for selected regions of interest in order to facilitate more detailed investigations on local scale.*

Line 155, w=3 for all the other regions shown in Figure 1?

**We applied w=3 for regions with strong seasonal flow variations and surging glaciers and w=1.5 as originally proposed by Luettig et al. (2017) for all other regions, for which a priori velocity information from ITS_LIVE is available. We added a column to Table S1 that contains the value of w, applied in each region.**

Line 174, add glacier outlines to distinguish the glacier and non-glacier region in Fig.2.4.6.8.9

**We thank the reviewer for his suggestion. When composing the figures, we also considered adding glacier outlines. However, we made a conscious decision not to do so, because the glacier outlines consist of many small, isolated parts over wide areas. Adding them substantially disturbs the appearance and the readability of the figures as it widely covers the velocity data (which is the main data of the figures). We therefore would like to leave the figures as they are. As an example, we added glacier outlines to Fig. 4 as suggested by the reviewer. We hope that from looking at the example, the reviewer understands our concerns.**

[Figure]

Line 278, Though color code of points in Figure 5 was shown in Fig.4, but it's hardly to know the location relative to glaciers (i.e., which point is at high elevation)

**We are sorry if the color coding caused some confusion. We added the following sentence to the captions of Fig. 4 and 5 and hope that this makes the position of the measuring points on the glacier clearer:**
*The color coding indicates the position of the corresponding measuring point on the glacier: purple=upstream. green=mid glacier, orange=front.*

Line 427 how about the accuracy of your annual mosaic products relative to ITS_LIVE product?

**In the text we discuss the differences between our annual mosaics and the ITS_LIVE products, as well as the possible reasons for the differences, in detail. One obvious difference between our product and the ITS_LIVE product with regard to accuracy are the blunders contained in the ITS-LIVE product in some featureless accumulation areas. However, while this is more a qualitative statement on accuracy, providing a general quantitative measure of relative accuracy for both data sets is very difficult, if not impossible. For such a comparison, one would actually need true mean annual velocity mosaics as reference, a data set that does not exist. Taking either our or the ITS_LIVE annual mosaics as "true" velocity reference is also not appropriate, because their calculated mean velocities depend only to a certain extent on the quality/accuracy of the input data (the blunders in the accumulation areas are an example). However, especially in the case of Svalbard with its strong seasonality of the glacier velocities and its surging glaciers with rapid speedups, most of the final mean velocity is determined by a) the temporal resolution of the input data (i.e. are short term velocity variations resolved and how many measurements of these variations go into the calculation of the mean?), b) the temporal coverage of the data (i.e. are there more velocity measurements during summer or winter, or are all seasons equally covered?) and c) the calculation of the mean itself (which weighting scheme is used, how are outliers determined,…). Nevertheless, as discussed in the text, our temporal mosaics come with several statistical layers that at least allow for a qualitative assessment of the velocity data by e.g. using standard errors along with the measurement count.**

Line 730, location of your example areas should be denoted.

**We added a note to the caption of Figure 1:**
*The example region Svalbard is marked with the number 7.*

Line 770, add some general statics of the difference

**We do not exactly know what the reviewer means with "statics" here and where to add them (to Figure 7, to its caption or to the main text). If the reviewer means "statistics", we would like to refer to the extensive statistical analysis of the velocity differences between TSX, S1 and Landsat 8 velocity fields in Sect. 3.2 and in Table 1.**